# A Feasibility Analysis of Energy Retrofit Initiatives Aimed at the Existing Property Assets Decarbonisation

Pierluigi Morano [1], Francesco Tajani [2], Felicia Di Liddo [1] and Paola Amoruso [3,*]

1 Department of Civil, Environmental, Land, Building Engineering and Chemistry, Polytechnic University of Bari, 70126 Bari, Italy; pierluigi.morano@poliba.it (P.M.); felicia.diliddo@poliba.it (F.D.L.)
2 Department of Architecture and Design, Sapienza University of Rome, 00196 Rome, Italy; francesco.tajani@uniroma1.it
3 Department of Engineering, LUM Giuseppe Degennaro University, 70010 Casamassima, Italy
* Correspondence: amoruso@lum.it

**Abstract:** In light of the growing demand for sustainability in the construction sector and real estate market, the European Community (EC) has recently begun incentivizing renovations of private and public European buildings. This aligns with the EC's aim to reduce harmful emissions by 55% by 2030 compared to 1990 levels, and to achieve complete decarbonization of buildings, i.e., zero harmful emissions in this sector, by 2050. Given this framework, this study aims to verify the financial viability associated with the construction of "green" buildings, as well as the associated monetary benefits related to the efficient nature of these buildings and the resulting reduction in energy consumption. Lastly, an investigation is conducted to determine the economic feasibility of energy retrofit initiatives on existing building assets by comparing the required costs to retrofit against the potential increase in market value of a retrofitted residential unit. Along with assessing the undisputed environmental advantages for the community and all building users, this research aims to assess the financial and economic feasibility of sustainable construction initiatives, providing insight into how best to pursue the EC's aims.

**Keywords:** decarbonization; energy retrofit; housing market; residential properties; financial feasibility; investment costs; management costs; market value; environmental benefits

## 1. Introduction

The transition towards a low-carbon construction sector represents one of the most significant and complex challenges in recent years. The construction industry plays a key role in contributing to climate change and the creation of a carbon neutral society; the sector is positioned to have considerable influence in addressing the push to limit the increase in average global temperatures to 1.5–2 °C compared to pre-industrial levels [1], as defined by the Paris Agreement [2,3].

The topic of decarbonisation in this field is extremely relevant, given that (i) in 2020 the construction sector was responsible for 37% of $CO_2$ emissions and 30% of global energy consumption [4]; (ii) currently, buildings account for 40% of final energy consumption in the European Union (EU), and 36% of its energy-related greenhouse gas emissions; and (iii) 75% of buildings in the EU are still energy inefficient.

In general terms, decarbonisation does not represent a simple or immediate solution to environmental issues. Solving such issues requires a complex process, necessitating a change of perspective to accommodate a systemic approach capable of reducing carbon emissions associated with the entire life cycle of a building. However, implementing decarbonisation measures can contribute to improving the well-being of building occupants, along with the creation of sustainable constructed environments.

Goals relating to the reduction of carbon emissions in the construction sector focus on the operational and embodied carbon of buildings, which represent different emission types related to different phases of the construction process. More specifically, the first component (operational carbon) refers to emissions of $CO_2$ and other greenhouse gases produced during the use and management of a building. This includes energy consumption associated with heating, cooling, lighting, household appliances, and other operating systems, all of which are influenced by occupant behaviour, and also encompasses the energy technologies used and the overall energy efficiency of the building. The second component (embodied carbon) is related to emissions of $CO_2$ and other greenhouse gases associated with the production, construction, and life cycle of materials used in the construction process, such as the extraction and processing of raw materials, the manufacture and transport of building materials, the construction of the building and, finally, its demolition and disposal [5–8].

Among the various decarbonisation tools and strategies to be implemented, the adoption of passive measures (such as opaque casings, including wool, cork, polyurethanes, transparent envelopes with PVC, aluminium, or wooden frames, and glass of various types) should by necessity be integrated with active renewable technologies in order to reduce energy consumption and the total environmental impact of buildings.

As part of the commitments undertaken as part of the Paris Agreement, the target of pursuing climate neutrality by 2050 set by the EU involves implementation of an economy with zero net greenhouse gas emissions [9,10]. In accordance with this goal, in 2018, the European Commission defined the "European strategic long-term vision for a prosperous, modern, competitive, and climate-neutral economy" [11], and has since launched various strategic initiatives, beginning with the Green Deal, in order to support the actual achievement of this important goal [12].

The global climate crisis has pushed EU nations to reconsider their current practices, especially in energy-intensive sectors like construction. In this sense, the EU has recognized the urgent need for decarbonisation. This has translated into the definition of initiatives aimed at the restoration of existing buildings and the construction of new green and zero-emission buildings.

The EU Green Homes Directive focuses on the improvement of the energy performance of buildings, with the aim of reducing harmful greenhouse gas emissions by 55% by 2030 (compared to 1990 levels), so as to ultimately achieve zero net emissions (in terms of economy-wide climate neutrality) by 2050. The directive specifically provides that all new buildings will be required to be carbon neutral (zero emissions) starting from 2028, whereas existing residential structures will need to achieve the energy label "E" by 1 January 2030, and the label "D" by 2033. There is also a provision for a ban on the use of fossil fuels by 2035, starting with the abolishing of subsidies for fossil fuel boilers by 2024 [13–15]. On an operational level, the EU member states should define not only any exemptions from the rule, but also all measures and incentives necessary to achieve the set targets. In particular, the individual states should be able to adapt their objectives based on the actual availability of qualified and skilled manpower, and the technical and economic feasibility of the renovation procedures. The final goal is to draw up a national restructuring plan capable of providing measures to facilitate access to targeted financing, along with a system of compensation for buildings requiring significant rehabilitation, and for needy families.

The transition to green buildings generates not only environmental benefits, but also considerable economic advantages [16,17]. Reductions in energy consumption translate into lower energy bills for residents and businesses, while the redevelopment of existing buildings and the construction of new buildings creates employment opportunities and stimulates innovation in the construction sector and in green technologies. Furthermore, low-energy and zero-emission buildings tend to have a higher market selling price, creating an incentive for investment in the real estate sector and implying a widespread increase in the overall value of existing building stock [18–21].

Improvements to public health are among the main advantages relating to the decarbonisation of the construction sector. In fact, green buildings improve the quality of internal and external air by reducing the presence of harmful substances, thus strengthening the life and work conditions of occupants. This aspect is particularly relevant in urban areas with high population and building density; such areas experience intensified environmental pollution and associated health effects [22,23].

In the global context, the aim for the construction sector is to reach zero emissions as soon as possible. This is to be accomplished through the creation of Nearly Zero Energy Buildings (NZEB, i.e., residential or non-residential buildings for which the annual balance between produced and consumed energy is close to zero). Therefore, building design should take into account a number of factors related to (i) exploitation of local resources and on-site energy production, (ii) the use of materials whose production does not cause high levels of carbon emissions, (iii) the optimization of consumption, and (iv) cutting energy waste while maintaining high qualitative standards.

Within the practices adopted for building decarbonisation, the implementation of laws on the use of recycled and renewable materials, the inclusion of solutions for improving energy performance, and the definition of incentives supporting the sustainability of building interventions are all fundamental in the establishment of strategic environmental initiatives. The integration of energy efficient solutions from renewable sources is consistent with the goal of promoting eco-design measures in all phases of the project life cycle, from the production and transport of building materials to the construction of the building and its maintenance up until eventual demolition. In all stages of the construction processes (extraction, transport, realization, management, and disposal), attention should be paid to the use of low-emission materials that are highly durable and do not require frequent maintenance and, if possible, also on the reuse of materials, which is preferable from a cyclical economic perspective.

## 2. Research Objectives

This study aims to examine the potential changes associated with energy retrofit interventions in existing buildings, with a special focus on the economic impact of these improvements (in terms of demolition/retrofit/reconstruction costs, operational cost reductions, and property value improvement). The analysis takes into account the entire cycle of construction, operational life, and decommissioning of buildings.

Starting from a case of demolition and new construction of a residential building through the use of innovative materials and techniques, we determine reductions in the consumption and related management costs for a potential user of the new property (named *innovative building*—energy label "A4") compared to a *standard* structure (i.e., characterized by a low energy label—"F"). This is followed by verification of the financial viability for potential investors to use their own capital and bear the risk of the investment in the demolition and construction of these buildings.

Our study aims to examine the relevant and current issue of the regeneration of existing assets, highlighting the fundamental role of evaluation in determining the associated benefits.

In particular, this study focuses on evaluating the triple feasibility of energy retrofit initiatives on residential properties (both in terms of realization ex novo and renovation of existing structures): (1) *Financial* viability, in terms of the demolition and reconstruction of the entire building, for the private operator who carries out the intervention; (2) *Monetary* convenience (referring to the single residential unit included in the building subject to intervention) for the user (occupant of the house), defined in terms of savings in property management, i.e., the cost to be borne for the use of the systems installed in domestic spaces; (3) *Economic* feasibility, defined in terms of (i) sustainability of the initial investment costs, and (ii) the possible increase in market value of the existing building stock, assessed as a proxy variable of the quality level of the referenced urban area. Specifically, the term "*financial*" refers to investment benefits for demolition and retrofit; the term

"*monetary*" refers to operational benefits (from energy savings); and the term "*economic*" is associated with property value improvement benefits (short and medium-term market value improvements).

In addition, this paper lists the environmental benefits that can be derived from the use of innovative and sustainable construction techniques through the reduction or complete cancellation of fossil fuel consumption and the limitation of carbon dioxide production, thereby making buildings self-sufficient.

Our study was carried out by comparing standard technologies and materials against innovative alternatives, such as high-performance envelopes and high-efficiency systems, highlighting the benefits in terms of (i) environmental sustainability, i.e., energy returns which are linked to lower management costs, and (ii) financial and economic feasibility, related to the profits expected by an investor who intends to develop "green" building interventions, as well as to the increase in the market value of real estate assets following redevelopment interventions.

This study thus intends to examine the pivotal role played by energy retrofit initiatives on existing properties as a response to the needs of communities, as well as to the legislative provisions and national and European objectives regarding the creation of sustainable and energy efficient cities. In this sense, the evaluation discipline provides valid support for decision-making processes aimed at defining both private and public building and urban transformation policies for implementation. The ever-increasing need to define tools capable of orienting choices towards aware and profitable actions for all interested parties highlights the importance of carrying out analysis of the interventions prior to their actual execution in order to (i) evaluate investment and management costs, (ii) identify expected increases in real estate value due to improvements in the performance of buildings, and (iii) determine the advantages for the natural environment, with a view to sustainable development. In this sense, our paper emphasizes the importance of sustainable design, in reference to both realization ex novo and the renovation of existing buildings. Our analysis compares (i) the investment cost of the new construction intervention (demolition and realization ex novo through the use of innovative materials and technologies for the construction of buildings characterized by the highest energy label—"A4"), (ii) the redevelopment cost that enables achievement of the minimum threshold foreseen by European provisions (i.e., energy label "E" by 1 January 2030), and (iii) the cost of the initiative aimed at energy retrofitting capable of obtaining energy label "A".

The structure of this study is organized as follows. In Section 3, analysis of the reference literature on the topic of the decarbonisation of buildings is carried out, highlighting the main aspects addressed in scientific contributions to this field. In Section 4, a case study related to the demolition and new construction of a building through the use of innovative techniques is presented, and the financial viability of the investment is verified. Subsequently, a comparison between energy consumption in a standard residential unit (energy label "F") and the same unit following demolition and reconstruction (innovative property, energy label "A4") is developed in order to verify the monetary convenience for the user. Subsequently, an analysis aimed at verifying the possible increase in the market value of more efficient properties from an energy point of view (class "A" or "E" from class "F") is carried out in order to verify the economic feasibility of upgrade initiatives for existing buildings. Finally, an analysis is conducted with the aim of estimating the investment costs needed for redevelopment intervention in standard real estate units in order to achieve energy labels "E" (minimum threshold according to legislative provisions) and "A", and comparisons are made with the new construction costs already estimated. A brief outline of the environmental benefits in terms of $CO_2$ emission reduction is also provided. In Section 5, the conclusions of the study are drawn, with potential future developments listed.

## 3. Background

Currently, there is a great deal of attention being paid to the issue of energy efficiency of existing building assets in the regulatory and political spheres. The scientific literature on the topic is in line with the importance of the question, enriched by numerous contributions aimed at investigating various aspects connected to the topic in diverse geographical contexts and with multiple purposes [24–27].

The large number of studies on these themes attests to the interest of academics in the theoretical in-depth study of energy matters, as well as to the need to define methodological and operational approaches and tools for practical support for the implementation of energy regeneration initiatives in buildings, in terms of both new developments and refurbishment interventions.

Among the relevant scientific papers to be included in this category, numerous studies have focused on the development of models for optimizing the energy services of residential buildings by improving thermal efficiency [28–32]. For example, Leibowicz et al. [33] developed an optimization model that incorporated three strategies for decarbonizing residential building energy services. These three strategies comprised (i) driving the transition to less carbon-intensive fuels in the energy supply mix of buildings, (ii) adopting more energy-efficient appliances and improving thermal insulation of buildings, and (iii) facilitating the decarbonisation process at the lowest cost.

The promotion of innovative projects in building energy codes was the key focus of a study carried out by Schwarz et al. in the context of Denmark, France, England, Switzerland, and Sweden. Their analysis developed six energy codes for buildings, which highlighted the promotion of decarbonisation through increasing energy efficiency and the use of renewable sources in order to accelerate the retrofit processes of existing building assets [34]. Similarly, with regards to cooling technologies and their suitability in providing comfortable buildings in hot and humid climates, Bandyopadhyay and Banerjee examined possible technological low-carbon solutions for space cooling, and identified pathways that could help reduce cooling loads and carbon emissions [35].

In addition, a multi-objective optimization approach was applied in [36], using dynamic energy simulation and life cycle assessment to minimize the life cycle global warming potential and related costs of a building and also investigate the sensitivity of environmentally optimal building design solutions to a variable electricity mix.

Sustainability goals and their close relationship with the construction sector were analysed by Jaglan and Korde, with a view to promoting a collaborative and synergistic approach in order to achieve a successful result. In [37], explanations were provided regarding the ways in which the construction sector affects greenhouse gas emissions, the main strategies to be adopted for decarbonisation, and the ways that parties involved could benefit from the process. In addition, Andrews and Jain investigated the effects of the implementation of various performance standards for commercial buildings in 15 cities in the U.S. from 2024 to 2050 in order to evaluate the potential reduction in greenhouse gas emissions resulting from different new building performance standards [38]. The issue of building decarbonization was also addressed by Naimoli, who outlined an overview of the potential for reducing operational greenhouse gas emissions related to the construction sector from a private point of view. The goal here was to identify and implement necessary solutions to lead the entire sector towards net zero emissions [39] during property creation and management, as well as in the provisioning of technology. Furthermore, this paper [40] discussed the most important near-term opportunities to advance building decarbonization, and identified five strategies to begin the transition to a low-carbon built environment, requiring the development of public policies and private investments to replace the use of fossil fuels for space and water heating and, in many cases, to improve the energy efficiency of buildings and reduce thermal losses.

In general terms, building-oriented decarbonisation initiatives deal with issues such as energy efficiency, on-site renewable energy generation, and integrated energy demand reduction measures [41]. Based on cases of low-carbon urban building governance in Stock-

holm, London, and San Francisco, along with interviews with experts, Tozer examined the effectiveness of policy initiatives in implementing urban decarbonisation processes. The author suggested solutions and operational approaches through which political mechanisms should allow these interventions to expand and become more durable over time [42].

Becqué et al. outlined eight pathways to decarbonize building stock through the achievement of Zero Carbon Buildings (ZCB). These buildings combine basic or advanced energy efficiency, on-site and/or off-site carbon-free renewable energy, and the use of carbon offsets (solely in cases where renewables cannot fully provide for 100% of the remaining energy demand). The work was based on reviews of current policy frameworks and consultations with stakeholders in four countries—India, China, Mexico and Kenya—to determine how policies at the national and subnational levels enable or disable the various ZCB components and pathways [43].

Research by Sofia et al. discussed the key role of mitigation policies in the progressive decarbonisation of the energy system in order to decelerate environmental pollution and generate social benefits (established by the National Energy and Climate Plan in Italy). This study aimed to evaluate how cost–benefit analysis could be implemented to quantify environmental and social costs and benefits in different sectors (energy, transport, and families), were the decarbonisation scenario to be applied in Italy in 2030 [44].

Also within the Italian territory, Tajani et al. proposed a model for evaluating the economic benefits, in terms of the feasibility of the operators involved, generated by energy retrofit interventions. Specifically, the authors tested the robustness of the innovative model in 110 provincial capitals by (i) determining the market value differential before and after energy intervention situations, and, assuming an ordinary profit margin of a generic investor interested in this type of investment, (ii) estimating the break-even incentive, i.e., the percentage threshold able to ensure the condition of minimum convenience for the investor [45].

The identification and subsequent quantification of the advantages connected to energy retrofit operations of buildings constitute two important objectives pursued in a number of studies carried out on the topic. Santamouris and Vasilakopoulou explored technological developments in the field of building energy to evaluate the current and future energetic, environmental, and economic potential of the field and illustrate diverse scenarios of potential energy consumption capable of favouring decarbonization of the sector, describing main advantages and critical issues [46].

Hopkins et al. examined policies to encourage energy efficiency, renewable energy, and clean transportation in California, and highlighted the financial, comfort, and health benefits for consumers. They provided policy recommendations to address the national building decarbonization challenge [47]. With the same goal in mind, Pachano et al. [48] addressed the issue of the electrification of building services through the replacement of fossil fuel use with renewable energy sources and the widespread installation of photovoltaic systems. The authors presented the application of a new rule-based control strategy to a case study through the implementation of a calibrated building energy model.

Gallagher and Holloway performed a United States-based systematic review of the integration of air quality and public health benefits deriving from decarbonization strategies by examining the current state of knowledge in the field and identifying opportunities for future policy action and further research [49]. Research developed by Ye et al. provided a perspective on the decarbonisation pathways of future buildings through a critical review of the existing scientific literature. They discussed the economic impacts of construction sector decarbonisation, including investments, environmental benefits, and changes in employment opportunities [50].

In China, a study carried out by Tang et al. developed a bottom-up energy technology model for the building sector to explore optimal paths to decarbonize the national building sector within global warming thresholds, as well as to evaluate the associated costs and benefits [51]. In addition, their study [52] examined the efforts of the Chinese construction sector to achieve $CO_2$ reduction targets. Furthermore, their research explored opportunities

leading to deep decarbonisation by considering urban and rural residential and services market segments. The researchers designed five different scenarios to evaluate the impacts of several energy and emission constraints in various situations in order to meet the goals set by the Paris Agreement.

The topic of cooperation among academics, stakeholders in the construction sector (clients, architects, engineers, producers, and financiers), and policy makers to promote the pursuit of strategic environmental objectives in the construction sector was addressed by Jaglan and Korde. They highlighted the actions that different parties can take in terms of future recommendations, including promotion of the reuse of components and the use of materials associated with lower $CO_2$ emissions [37].

The present study is linked to the issue of sustainability in the construction sector as it describes a case study that involves a demolition and new construction intervention, as well as an energy retrofit project, all for the same building. With reference to the initiatives under study, the financial, economic, monetary, and environmental benefits connected to their implementation are assessed in order to analyse the sustainability of the studied operations from diverse points of view.

## 4. Case Study

### 4.1. Analysis of a Demolition and New Construction of a Residential Building

An analysis was carried out to verify financial viability for a party investing in the demolition of an obsolete building and new construction of a "green" *innovative building*. The analysed building is part of a residential complex located in a suburban area of a Southern Italian city in the Apulia region. Specifically, the complex consists of eight buildings, all similar in construction characteristics, outline, and type. Initially, a single building stood in one part of the area; this building was constructed prior to 1976, and therefore did not meet the minimum regulatory requirements for energy efficiency that were subsequently introduced. In 2010, a construction company, which already owned the site, initiated an operation to demolish the existing building and construct a residential complex of eight buildings with similar design and construction characteristics. The construction phase of the complex was set in progressive time steps, involving the construction of one building at a time; each building took about 18 months to complete.

This case study is based on the analysis of a sample building. The new construction initiative was started in the second half of 2020 with the urban planning of the investment and the demolition of the pre-existing building. The construction of the new building was then initiated in 2021. This new building consists of eighteen building units ranging in size from 75 to 193 m$^2$. Five different floor plan solutions were designed in terms of room layout, distributed over five above-ground levels, with a total area of approximately 1930 m$^2$.

According to the information obtained directly from the construction company, the building has a structure consisting of reinforced concrete beams and pillars with latero-concrete floors. The foundations are made of concrete screed and plinths connected by concrete beams, while the roof consists of a concrete mix screed and layered insulation materials. The floors have a mixed brick and reinforced concrete structure with precast reinforced concrete joists, while the interior partitions are made of gypsum-fibre panels, galvanized steel, and fiberglass insulation.

In terms of systems, each housing unit has the following plant equipment:

- 11 kW heat pump system for winter–summer air conditioning;
- Ceiling heating and cooling by radiant panel with 8 mm piping inside, insulated with EPS200 expanded polyester;
- 300 L energy efficient, class "B" electric boiler for domestic hot water production;
- Controlled dehumidification system for air treatment with air exchange in the various rooms, equipped with a heat recovery unit with more than 90% efficiency;
- Controlled mechanical ventilation (VMC) that allows continuous air exchange, ensuring a healthier and more comfortable environment;

- Photovoltaic system with polycrystalline silicon panels, providing semi-autonomy for the housing units.

Verification of the Financial Viability of the Intervention

In the first instance, a financial analysis of the building intervention was carried out, aimed at verifying the viability of the initiative for the developer based on the profitability of the initiative. To develop this analysis, data provided by the company regarding the intervention costs, the construction time, the sale price of the realized housing units, and the energy consumption of the systems per apartment were used.

Specifically, the total costs for the implementation of the project and for the completion of the building intervention (demolition, disposal, and construction of the sample building) amounted to € 2,509,000, temporally distributed as shown in the cost distribution table (Table 1), which when related to the total area of the building, i.e., 1930 m$^2$, resulted in a unit construction cost of € 1300/m$^2$. The company reported that it raised part of the necessary capital through a bank loan, whose mortgage repayment plan is shown in Table 2, with a 2-year term, for a total financed amount of € 2,000,000 at an annual interest rate of 2.0%.

**Table 1.** Percentage incidence of construction costs.

| Num | Activity | Total Cost | % Incidence |
|---|---|---|---|
| 1 | Design | € 75,270.00 | 3.00% |
| 2 | Construction site and excavation | € 150,540.00 | 6.00% |
| 3 | Foundation structures | € 137,995.00 | 5.50% |
| 4 | Provisional works | € 37,635.00 | 1.50% |
| 5 | Concrete structure | € 840,515.00 | 33.50% |
| 6 | Roofing | € 188,175.00 | 7.50% |
| 7 | Masonry | € 363,805.00 | 14.50% |
| 8 | Fixtures and screeds | € 476,710.00 | 19.00% |
| 9 | Flooring and wall coverings | € 87,815.00 | 3.50% |
| 10 | Plastering and painting | € 62,725.00 | 2.50% |
| 11 | Plumbing installation and finishing | € 50,180.00 | 2.00% |
| 12 | Site dismantling | € 37,635.00 | 1.50% |
| | **Total construction cost of the building** | € 2,509,000.00 | 100.00% |

**Table 2.** Mortgage repayment plan.

| Year | Installment | Principal Rate | Interest Rate | Outstanding Debt |
|---|---|---|---|---|
| 0 | | | | € 2,000,000.00 |
| 1 | € 1,030,099.01 | € 999,099.01 | € 40,000.00 | € 1,009,900.99 |
| 2 | € 1,030,099.01 | € 1,009,900.99 | € 20,198.02 | € 0.00 |
| Capital financed | € 2,000,000.00 | | | |
| Years | 2 | | | |
| Rate | 2.0% | | | |

The financial analysis of the operation was carried out through the application of a *Discounted Cash Flow Analysis* (DCFA).

Regarding the reference time overview, the investment started in 2020 with a duration of 18 months. To estimate the investment and operating costs, a total intervention cost of approximately € 2,509,000 (including design costs, assumed to be 3.0% of the construction

cost reported by the company) was assumed, while the cost of debt capital amounted to € 60,198.02, as calculated in the mortgage repayment plan (Table 2), to be added to the total intervention cost. Since the developer already owned the area, there were no acquisition costs incurred.

The second stage of the DCFA focused on the determination of revenues from the sale of the realized units. The company, due to the high construction quality of the buildings and their technical characteristics, reported that it sold the residential units at an average price of € 2000/m², which, multiplied by the total building area of 1930 m², resulted in total sales revenue of € 3,860,000. Next, the time point at which each cost and/or revenue return occurred was identified. The company reported that for all properties, there were off-plan sales during the first year of the intervention. Payments due from the buyers were made at regular intervals throughout the period of the intervention, at the sixth, twelfth, and eighteenth months, respectively, in addition to an initial down payment, which, like the instalments at six and twelve months, was 20% of the property's sale price; at the eighteenth month, the buyers notarized paying the remaining 40% of the total property price. Thus, revenues were recorded by the company according to the timeline shown in the Sales Plan (Table 3).

**Table 3.** Sales Plan.

|  | Down Payment | I Share | II Share | III Share |
|---|---|---|---|---|
| Month | 0 | 6 | 12 | 18 |
| Share % | 20% | 20% | 20% | 40% |
| Revenues | € 772,000.00 | € 772,000.00€ | € 772,000.00 | € 1,544,000.00 |
| TOTAL | € 772,000.00 | € 1,544,000.00 | € 2,316,000.00 | € 3,860,000.00 |
| **Total sales revenue** | **€ 3,860,000.00** | | | |

The unit sales price was higher than the average zone unit value for civilian housing with standard construction characteristics and average plant equipment. The aforementioned zone unit value of € 1325 /m² was inferred as the average of the unit values, indicated by the Observatory of the Real Estate Market (OMI) of the Inland Revenue Agency [53]. This database identifies a range of unit values for properties in the specific micro-zones into which municipal territories are divided, on the basis of the purchase and sale prices reported in notarial deeds. The range of values provided by the OMI for the reference zone and relating to the first half of the year 2022 varied from € 1200/m² to € 1450/m². It is evident that the final sale price of the newly built properties under consideration was higher than the maximum value identified by the OMI. This is explained by the fact that the database generically refers to a normal state of preservation, while in the case under consideration, we were dealing with above-average properties in terms of plant equipment and construction characteristics.

The difference in sale prices between the analysed units and the area average was therefore attributable to the innovative construction techniques used, the quality of the materials, and the level of comfort of the units, aspects that grant the units greater standards of quality, sustainability, and liveability than the average properties offered in the area, which are characterized by a more advanced degree of obsolescence. This circumstance therefore resulted in an increase in market value for the units of about 50.95% compared to the area average.

After determining the cash flows generated by the investment over the reference time horizon, it was necessary to discount them to the present. This operation allowed the positive and negative amounts generated by the initiative over its time horizon to be

carried forward to the same valuation point. Specifically, the discounting operation was based on the following formula:

$$F = \frac{1}{(1+i)^t} = \frac{1}{q^t}$$

where "*i*" denotes the discount rate, i.e., the opportunity cost of capital for the operator, or the best expected return by comparing it to other alternative forms of investment with similar riskiness that the investor has given up in order to allocate financial resources to the project in question, and "*t*" denotes the time point at which the cash flow occurs.

The discount rate calculated by the direct procedure was given by the sum of three components:

1.  The guaranteed return that the investor would obtain by deploying the capital in alternative risk-free assets. The average annual rate of return on government bonds of similar duration to that of the investment was taken as a reference. According to the 2021–2022 Bank of Italy's Rendistato, for Treasury Bonds with a duration of 12 to 18 months, the return averaged 3.6% [54];
2.  Expected inflation (medium- to long-term expected by the financial market). This, for the investment term, was 5.0%, obtained as the average of values for the years 2021 and 2022, as reported by National Institute of Statistics (ISTAT) [55];
3.  The additional premium expected by the investor for the risk incurred with the intervention (depends on the investor's risk appetite, the generic risk of the sector to which the investment belongs, and the specific risk of the investment). In this case, since the project was characterized as low risk, the premium was assumed to be 2.0%;

    Hence, the calculated discount rate was 10.6%.

Subsequently, financial profitability indicators were determined to verify the financial viability of the investment. These included the Net Present Value (NPV), which is given by the discounted sum of cash flows. It represents the incremental wealth, in monetary terms, eventually generated by the initiative. If NPV is greater than 0, the investment generates some incremental wealth.

Once the necessary data were determined, the DCFA was developed, the results of which are shown below (Table 4).

**Table 4.** DCFA development.

|  | Urban Planning Process, Demolition and Construction | Completion and Sale of Housing Units |
|---|---|---|
| **Period** | **0–12 months** | **13–18 months** |
| **Construction Costs** | | |
| building | € 1,756,300.00 | € 752,700.00 |
| **General Costs** | | |
| interest expenses | € 40,000.00 | € 20,198.02 |
| **Total Investment Costs** | € 1,796,300.00 | € 772,898.02 |
| sales | € 2,316,000.00 | € 1,544,000.00 |
| **Total Revenues** | € 2,316,000.00 | € 1,544,000.00 |
| **Cash Flow** | € 519,700.00 | € 771,101.98 |
| **NPV** | **€ 1,100,270.25** | |

In light of the determined NPV value, the financial viability of the initiative was tested.

*4.2. Verification of the Monetary Convenience of the Typical Real Estate Unit: Estimation of Operating Costs*

In this Section, an analysis was carried out to identify the monetary convenience, in terms of management cost savings, associated with a newly constructed "*innovative*" property (energy class "A4") compared to a "*standard*" property (energy class "F"), with reference to a 75 m$^2$ typical real estate unit.

4.2.1. Estimated Consumption of the "Innovative" Property and the "Standard" Property

Regarding the energy consumption of individual building units, the energy analysis was developed using the "Termus" software (v.40), assuming a typical building unit with a total gross floor area of 75 m$^2$ as the case study.

For a newly built property, based on the consumption reported by the construction company, this analysis returned a total global non-renewable primary energy consumption (EP$_{gl,nren}$) of about 10.24 kWh/m$^2$ per year, a global renewable primary energy consumption of 50.05 kWh/m$^2$ per year, and an estimated CO$_2$ production rate of 4.48 kg/m$^2$ per year. Specifically, the annual electricity demand was 1886 kWh, of which 680 kWh was from the national grid, and 1186 kWh was supplied by the photovoltaic system. The only energy used is electricity due to the installation of heat pumps, which are more efficient than traditional condensing boilers. In addition, many of the energy needs are met by the photovoltaic system, which provides additional monetary savings. It follows that in the calculation of costs associated with consumption, the share of electricity supplied by the photovoltaic system was not considered.

Therefore, in order to proceed with the calculation of costs on the bill, the unit cost of electricity must be determined and multiplied by the amount required by the "*innovative*" property. The regulated market price proposed by ARERA (*Regulatory Authority for Energy Networks and Environment*), which adjusts the economic and contractual conditions quarterly according to changes in the market price of raw material, was used [56]. Thus, with regard to the total annual cost of electricity for a single-hourly property with a power demand of 3 kW, referring to the second quarter of 2023, three elements were referred to:

- Energy share, including charges, equal to € 0.15664/kWh;
- Fixed fee equal to € 79.0403/year;
- Power quota equal to € 1.71/kWh.

Thus, the "*innovative*" housing unit was estimated to have an annual operating cost, related to electricity consumption, of € 1348.35/year.

For the monetary evaluation of the consumption of the "*standard*" building (class "F"), we used the results obtained through the energy analysis implemented using the "Termus" software. For the "*standard*" type building (class "F"), the annual electricity needs in standard use are equal to 671.64 kWh per year, and the annual natural gas needs are equal to 1378.31 sm$^3$ per year. Thus, the electricity costs of the reference building unit in energy class "F" with a consumption of 671.64 kWh per year were estimated at € 1331.55/year, calculated by multiplying this consumption expressed in kWh by the cost of electricity, specified above. As for the cost of methane in the reference housing unit (class "F"), located in Apulia and with consumption in the range 481–1560 sm$^3$, this is calculated using:

- An energy fee equal to € 0.631268/sm$^3$;
- A fixed fee equal to € 96.95/year.

Thus, the total methane cost for the reference housing unit in energy class "F" with a consumption of 1378.31 sm$^3$ per year was calculated at € 967.03/year. It follows that the total annual operating costs of the energy class "F" property were projected to be € 2298.58/year.

### 4.2.2. Comparison of Consumption of the Two Types of Properties ("Standard" vs. "Innovative")

In this Section, two building types "*standard*" (class "F") and "*innovative*" (class "A4") were compared. Specifically, the average annual consumption of both solutions was compared, and an indication of their monetary cost was provided.

The "*innovative*" type of building, as shown above, had no consumption related to methane fuel and required only the use of electricity, amounting to 1886 kWh, including 680 kWh from the national grid and 1186 kWh from the photovoltaic system. This resulted in an annual operating cost, due to electricity consumption, of € 1348.35/year.

In contrast, the consumption expenditure attributable to the property in energy class "F", amounting to € 2298.58/year, was almost double that associated with the similar unit with energy class "A4". In particular, the A4 unit exhibited a percentage decrease of −41.34% in energy consumption, representing a total net saving of € 950.23/year.

### 4.3. *Analysis of the Economic Feasibility of Upgrading*

We first determined the economic impact of a hypothetical upgrading intervention on an existing building (class "F") to achieve class "E" (minimum limit provided by the EU) and class "A", respectively. These costs were compared with the unit cost associated with construction of the new "*innovative*" property (class "A4"), as reported by the company. To verify the cost-effectiveness associated with upgrading interventions, the possible increase in the market value of the existing property, ante- (class "F") and post (class "A" and class "E")-upgrading intervention, was then estimated. The evaluation of the market value was exclusively used to verify the economic feasibility of the investment in terms of increase in the value of the property compared to the necessary costs, without any prejudice to the certain environmental benefits connected to the decarbonization of buildings.

### 4.3.1. Estimation of Redevelopment Costs and Comparison with That of New Construction

With regard to initial costs, two estimation criteria can generally be used: (i) Bill of quantities, which consists of analysing the individual components that make up the property in order to develop the "quantities" (in terms of surface area/volume) of each to be reported in the metric calculation. Then, these quantities are multiplied by the unit price of each material and workmanship, which can be deduced from official price lists such as, for example, the *Regional Price List of Public Works of Apulia* or the *DEI Price List for materials and completed works*. Finally, these amounts are added together, determining the total cost of the work; (ii) The summary estimate using the comparative method, which is used by taking as a reference the total unit construction cost referring to the same type of construction inferred from similar interventions carried out in the same geographical area and with similar workmanship or, alternatively, by referring to official price lists by building type (e.g., *DEI price list by building type*).

The initial intervention costs, i.e., the total outlay for the rehabilitation of the "*standard*" building versus the total cost of construction of the "*innovative*" one, are analysed below. In the present case, the information provided by the construction company was used to determine the intervention costs of the "*innovative*" building. For the estimation of the costs related to upgrading of the "*standard*" building to achieve energy class "A", we used two sources:

- Dlgs. of 14/02/2022 (*Definition of specific maximum costs that can be facilitated, for certain types of goods, within the framework of tax deductions for buildings*) [57], published in the Official Gazette of the Republic of Italy, by which the Ministry of Ecological Transition established maximum prices for carrying out specific upgrading interventions aimed at increasing the energy efficiency of Italian real estate stock. The main interventions, with the corresponding maximum expenditure amounts provided, were as follows:

    (a) roof insulation by affixing insulating panels to be applied to the outside of the roof to limit heat loss: maximum € 276/m$^2$;

(b) insulation of floors using a layer of insulating material laid on the outer surface, to prevent heat loss through the ground: maximum € 144/m$^2$;

(c) insulation of perimeter walls by insufflation of insulating material or using panels with low thermal conductivity: maximum € 195/m$^2$;

(d) replacement of window frames with thermal break type and double/triple insulated glass: maximum € 780/m$^2$;

(e) replacement of obsolete systems with high-efficiency condensing systems, associated with floor or wall systems: maximum € 240/kW;

(f) domestic hot water production systems with heat pump water heaters: maximum € 1500.

- The upgrading interventions outlined in Directive 2010/31/EU art. 5 [58], issued by Italy in response to requests from the European Community, were assumed essential, namely: thermal insulation of the building envelope (roofing slab, opaque perimeter walls dispersing and reduction of thermal bridges); replacement of windows and doors; replacement of the heat generator; and the use of renewable sources (solar thermal, photovoltaic).

Thus, related to the interventions of envelope efficiency and replacement of windows and doors, a maximum expenditure of € 1395/m$^2$ was determined. Regarding the replacement of the heat generator, the maximum expenditure corresponded to € 240/kW, from which followed a total cost to upgrade the reference real estate unit (75 m$^2$) of € 108,225 (considering a 15 kW condensing boiler), or € 1443/m$^2$, compared to the cost of construction associated with building a new "*innovative*" property of € 97,500. This was obtained by multiplying the unit construction cost reported by the company, equal to 1300.00 €/m$^2$, by the gross area of 75 m$^2$. The upgrading interventions considered would bring a significant improvement in terms of energy efficiency and non-renewable primary energy consumption, thus allowing the transition from energy class "F" to a class "A"; however, the cost of upgrade was found to be 11% higher than the cost of construction to build a new "*innovative*" property.

Subsequently, a second survey was conducted to estimate the retraining costs necessary to achieve the minimum energy class required by the European Community (class "E") based on the document issued by the Ministry of Economic Development in collaboration with the Ministry of the Environment and Protection of Land and Sea and the Ministry of Infrastructure and Transport in 2020 [59]. This document expresses the overall average cost of upgrading a property with the goal of achieving the minimum energy class defined by Europe in the Energy Performance of Buildings Directive (EPBD) of 14 March 2023, called the Green Homes Directive [15]. Specifically, this directive states that residential buildings should have a minimum energy class of "E" by 2030, and "D" by 2033. As a result, for a property in climate zone D, the same as the reference building unit, and which has not undergone renovation, the minimum overall cost of upgrading is € 355/m$^2$ (*Table 20 Strategy for the Energy Upgrading of the National Housing Stock*). This is considered as the minimum threshold value, to which the cost of a 15 kW condensing boiler, amounting to about € 3600, should be added. As previously clarified, this amount relates to an intervention aimed at reducing the consumption of non-renewable primary energy up to a value of 101 kWh/m$^2$, corresponding to energy class "E", the threshold limit provided by European regulations. Thus, proceeding as in the previous case, the estimated cost of upgrading the reference building unit (75 m$^2$) to achieve class "E" would be € 30,225, significantly lower than both the cost of building the new "*innovative*" building and the cost of upgrading to achieve class "A". It should be highlighted that the considered construction costs for energy retrofits do not include demolition or removal costs and expenses related to technical aspects.

4.3.2. Change in Market Value of the Property before and after Upgrading

To verify the change in the market value of the "*standard*" property before (class "F") and after (class "A" and class "E") the redevelopment intervention, we first estimated the market value of the property characterized by a certain level of technical–constructive

obsolescence, and not having undergone recent redevelopment interventions; then, we compared it to the market value of properties that had undergone contained redevelopment interventions (representative of class "E"), as well as properties that had been subject to more substantial interventions (representative of class "A"). The market values were determined on the basis of information provided by the main market players, such as real estate buying and selling portals and real estate operators. In order to determine the market value of the property prior to the redevelopment work, we made use of the "Borsino Pro" portal, a real estate valuation platform that returns the probable market value of property using data inferred from notarized deeds. This value was compared with what was determined via analysing comparable properties, i.e., properties with similar characteristics to the property to be valued (location, finishes, maintenance status, etc.). Once the comparable properties were identified, their unit value ($€/m^2$) was determined, and the arithmetic mean of the sample was determined. The first survey, conducted with the "Borsino Pro" portal, returned a unit value of $€\,1102/m^2$ for an apartment matching typical type and construction characteristics for the area. For the second analysis, the following characteristics were used in the selection of comparable properties: the presence of three rooms; an area between 60 and 110 $m^2$; intermediate floor level; not renovated with a condensing central heating system; and location in the same area.

This estimation procedure, comparing the unit prices of six properties homogeneous with the unit under study, resulted in a unit value of $€\,1138.75/m^2$. Since the comparable properties were represented by sales offers, a 10% deduction was applied as an ordinary discount coefficient between sales offer and final price. This percentage value was obtained by consulting the main operators of the local real estate market. The unit values determined by the two analyses were close to each other, and it was determined that the reference value would be that determined by the analysis of the comparable properties, as this value's reliability was validated by the "Borsino Pro" platform. This value, $€\,1138.75/m^2$, was approximated to $€\,1140.00/m^2$.

Using a similar procedure for assets that had undergone lesser (class "E") and substantial (class "A") redevelopment, a market value of $€\,1309/m^2$ was arrived at using the "Borsino Pro" platform. However, this value was considered unreliable, because the platform did not allow for the energy class and reference plant equipment to be taken into account. In order to overcome this limitation, reference was made exclusively to the analysis conducted using comparable properties, which led to the determination of a unit value of $€\,1960.20/m^2$ for class "A" properties, which can be approximated to $€\,1960.00/m^2$, and 1352.28 $€/m^2$ for class "E" properties, which can be approximated to $€\,1350.00/m^2$. Table 5 summarizes the comparable properties used in the three estimates conducted.

**Table 5.** Comparable properties.

| Sample | Standard Property | | | Upgraded Property Class "A" | | | Upgraded Property Class "E" | | |
|---|---|---|---|---|---|---|---|---|---|
| | Area (m$^2$) | Price (€) | Unit Price (€/m$^2$) | Area (m$^2$) | Price (€) | Unit Price (€/m$^2$) | Area (m$^2$) | Price (€) | Unit Price (€/m$^2$) |
| 1 | 60 | 59,000.00 | 988.33 | 105 | 205,000.00 | 1952.38 | 83 | 100,000.00 | 1204.819 |
| 2 | 85 | 110,000.00 | 1294.12 | 96 | 198,000.00 | 2062.50 | 115 | 164,000.00 | 1426.087 |
| 3 | 110 | 125,000.00 | 1136.36 | 85 | 155,000.00 | 1823.53 | 110 | 153,000.00 | 1390.909 |
| 4 | 105 | 120,000.00 | 1142.86 | 110 | 199,000.00 | 1809.09 | 119 | 168,000.00 | 1411.765 |
| 5 | 73 | 69,000.00 | 945.21 | 102 | 185,000.00 | 1813.73 | 95 | 129,000.00 | 1357.895 |
| 6 | 93 | 123,750.00 | 1330.65 | 100 | 230,000.00 | 2300.00 | 90 | 119,000.00 | 1322.222 |
| **Mean Value** | | | **1138.75** | | | **1960.20** | | | **1352.28** |

As might be expected, the market value of the properties characterized by more obsolete construction and plant engineering techniques was lower than that of the properties

that had been upgraded and thus were more energy efficient. In fact, there was an increase in market value of about 71.92% for class "A" properties, which corresponded in absolute value to a difference of € 830/m². For class "E" properties, there was an increase of about 18.5%, or € 210/m², which although more limited than the class "A" properties nevertheless denotes appreciation in the market. However, this difference in value is lower than the minimum overall cost of upgrading provided in the Case Green Directive for achieving class "E", which is € 403/m²; similarly, the increase in market value recorded for class "A" properties (€ 830 /m²) is also lower than the estimated cost incidence for achieving this class (€ 1443/m²). It has also been clarified that the cost estimate does not take into account demolition or removal costs and expenses related to technical aspects.

It follows that energy upgrading interventions, with the aim of achieving the minimum performance required by the EU (class "E") (or higher classes), are not convenient from an economic point of view, despite the fact that these upgrades are convenient from a monetary point of view, in terms of consumption reduction. This economic imbalance represents the need for an adequate system of incentives to support the economic sustainability of this type of initiative.

Table 6 summarizes the data and results achieved with the financial analyses: the monetary analysis of consumption, the economic analysis of the intervention costs (redevelopment and new construction), and the economic analysis of the market values of the "*innovative*" property (corresponding to the sale price of the "*innovative*" property of new construction) and the existing property (estimated by applying the comparative method for a *standard* property before and after redevelopment).

**Table 6.** Summary of results.

| | "Standard" Existing Property | Redevelopment (from Energy Class "F") | | New Construction |
|---|---|---|---|---|
| Energy class | Class "F" | Class "E" | Class "A" | Class "A4" |
| Initial costs (€/m²) | | 403.00 | 1443.00 | 1300.00 |
| Monetary convenience (€/year) | 2298.58 | | | 1348.35 |
| Market value (€/m²) | 1140.00 | 1350.00 | 1960.00 | 2000.00 |
| ENVIRONMENTAL BENEFITS | | | | |
| $CO_2$ produced (kg/m²) | 39.35 | | | 4.48 |
| EPgl,nren (kWh/m²) | 207.94 | | | 10.24 |
| EPgl,ren (kWh/m²) | 0 | | | 50.05 |

Environmental benefits are understood not only as benefits brought to the environment, but also to individuals. They are related to the use of innovative, efficient, and environmentally sustainable technologies, plants, and materials, and the reduction of $CO_2$ emissions is a priority associated objective. Increased energy efficiency enables a significant reduction in the use of fossil fuels and thus in the level of greenhouse gas emissions, which contribute to global warming. Energy efficiency, combined with energy production systems from renewable sources, enables faster achievement of the environmental goals set by Europe. Italy's housing stock is characterized by a high level of "historicity", characterized by the widespread presence of often obsolete technological systems, a circumstance that requires substantial investment to overcome in order to achieve the goals set by the EU. The pressing need for energy efficiency and environmental compliance in the housing sector must be taken into account.

## 5. Conclusions

This paper has investigated several aspects of financial, monetary, and economic feasibility related to the energy efficiency of buildings, both new and as part of the existing real estate stock.

Firstly, the financial viability for those investing in initiatives aimed at the construction of "green" buildings, designed through adapting buildings to climate change, including aspects that concern the life cycle of materials and the quality of interior spaces, has been verified. In this case, a precise orientation of investment choices is required, starting from the design phase, against a verified economic return for investors as well.

In implementing the most recent intentions of the European Community, which, with the EPBD, aims to upgrade the European building stock by improving its energy efficiency, further analyses were carried out. First, differences in monetary convenience, in terms of consumption reduction, between a "*standard*" building (class "F") and an "*innovative*" building (class "A4") were verified. Then, the economic feasibility associated with upgrading interventions of a "*standard*" property to class "F", to reach class "E" (minimum limit provided by the EU) and class "A", respectively, was assessed; these costs were compared with the cost of building an *innovative* property (class "A4") to new. Finally, the economic feasibility of upgrading interventions was verified by determining the market value of the properties before (class "F") and after (class "E" and class "A") these interventions, thus estimating their possible increase and comparing it to the required intervention costs.

This survey is particularly topical given the recent policies of the EU, which have set the goal of incentivizing renovations of European private and public buildings and reducing harmful emissions by 55 percent by 2030 compared to 1990 levels, and achieving full decarbonization of buildings by 2050, with zero harmful emissions. It is in fact planned to upgrade most residential buildings to class "E" by 2030, and then to class "D" by 2033.

Therefore, the costs of upgrading an existing class "F" property to achieve class "E" and class "A" were analysed and compared with the cost of building a class "A4" property new. It was verified that energy upgrades, with the goal of attaining the minimum performance standards required by the EU (class "E"), have a significantly lower economic impact than the charge required for upgrading to class "A", while still providing better energy performance; on the other hand, better energy efficiency was found to be characterized by higher market value appreciation. However, the increase in market value was found to not be sufficiently substantial enough to ensure the cost-effectiveness of the upgrading intervention, since for both the upgrade to class "E" and the upgrade to class "A", the increase in market value was less than the costs associated with upgrading. The achievable environmental benefits were also confirmed by a reduction in the amount of $CO_2$ produced.

The results achieved confirm the uncertainties raised in the Italian context in the face of EU demands. In fact, without prejudice to the undisputed environmental and savings advantages for users, if one considers the "historicity" of Italy's real estate heritage and the huge investments needed for existing buildings, many of which were constructed prior to 1976, the year in which the minimum regulatory requirements for energy efficiency were introduced, a realistic concretization of the objectives set by the EU should be supported by an adequate system of incentives, which could also include reorganization of the fragmented system of tax deductions. While the pressing need for energy efficiency and environmental compliance in the real estate sector is reflected in terms of financial and monetary convenience for those involved, a realistic achievement of the European objectives would require adjustment of the general forecasts of the European Community to the historical-cultural and climatic peculiarities of the different countries of the Union, alongside a system of incentives aimed at ensuring adequate economic support and concreteness in the face of the specific characteristics of differing geographical areas. The insights provided by this research may prompt a detailed discussion on the nature and impact of these incentive programs. In addition, this research can be adapted to different European territories, as the proposed evaluation techniques are replicable in various contexts and can be aligned with the peculiarities (i.e., climatic, legislative, historicity of real estate heritage) of specific territories. Finally, other future developments relating to this research could include new insights on energy consumption differentials for all of the various scenarios considered (e.g., class F, class E after renovation, class A after energy retrofit, and new construction).

**Author Contributions:** Conceptualization, P.M., F.T., F.D.L. and P.A.; methodology, F.D.L. and P.A.; validation, P.M. and F.T.; formal analysis, F.D.L. and P.A.; investigation, F.D.L. and P.A.; data curation, P.M., F.T., F.D.L. and P.A.; writing—original draft preparation, F.D.L. and P.A.; supervision, P.M. and F.T. All authors have read and agreed to the published version of the manuscript.

**Funding:** The research has been developed within the project "MISTRAL—a toolkit for dynaMic health Impact analysiS to predicT disability-Related costs in the Aging population based on three case studies of steeL-industry exposed areas in Europe"-HORIZON-HLTH-2022-ENVHLTH04—Grant Agreement Project n. 101095119 of the Polytechnic University of Bari (Italy).

**Institutional Review Board Statement:** Not applicable.

**Informed Consent Statement:** Not applicable.

**Data Availability Statement:** The data is contained within the article.

**Conflicts of Interest:** The authors declare no conflict of interest.

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
