# Peer review of "A Feasibility Analysis of Energy Retrofit Initiatives Aimed at the Existing Property Assets Decarbonisation"

_sustainability, doi:10.3390/su16083204_

Round 1

Reviewer 1 Report

Comments and Suggestions for Authors

The paper, titled "A Feasibility Analysis of Energy Retrofit Initiatives Aimed at the Existing Property Assets Decarbonisation," sets out to investigate the financial viability and economic feasibility of energy retrofitting in the construction sector, in alignment with European Union targets for reducing emissions and achieving a carbon-neutral economy by 2050. The manuscript is well-written. Below are my comments, which I believe will strengthen the impact and clarity of this research.

This paper limits its analysis to comparing the market value of specific architectural projects with that of unrenovated, outdated buildings, which does not sufficiently represent the economic advantages of newly constructed decarbonized buildings. The reliance on the concept of "market price" fails to effectively demonstrate the economic benefits of new decarbonized construction. This is because any newly renovated building, even without green certification, is likely to have a higher market value than an outdated building. Furthermore, the paper lacks sufficient discussion on certain topics. For example, it only briefly mentions that the economic benefits of reconstruct old buildings into more energy-efficient ones are not significant and suggests the need for incentive programs. However, it does not explore or analyze what these incentive programs might entail or their potential effectiveness. I believe that a detailed discussion on the nature and impact of such incentive programs would significantly enhance the value of this research.  Lastly, the literature review presented in the background section appears somewhat disorganized and could benefit from further categorization.

Reviewer 2 Report

Comments and Suggestions for Authors

The paper is of interest considering the need to retrofit existing building stock in Europe. This is especially relevant as the current retrofit rates are much lower than desired.

However, there is much to be clarified/corrected in this paper.

English used for the paper needs urgent reworking, especially, Section 1-3. There are multiple issues with the English used:

1. It almost feels like somebody has used a thesaurus in a haphazard fashion in places. Either that or the translation from the authors' native language to English has led to it looking like that.

2. There are entire paragraphs which are a single sentence broken up with a comma or two. This makes these sections nigh impossible to comprehend. (e.g., the entire paragraph that covers lines 40-45 in the manuscript is one big sentence)

Also, clarify your assumptions and definitions, e.g., In Line 110-111, NZEB is defined without bringing in the duration aspect (without mentioning duration, Net Zero makes no sense, is it net zero energy over a year, or is it a month, or a week?)

A major source of confusion in the paper is the use of the words: financial, monetary and economic. In English, economic, financial and monetary are used synonymously. So, the usage "financial, monetary and economical perspective/convenience", seems highly tautological.

It would be better to clarify the usage, either by rephrasing along the lines of investment benefits (for demolition and retrofit), operational benefits (from energy savings) and property value improvement benefits (short and medium-term market value improvements) OR some other equivalent terms.

This would greatly improve comprehension and get rid of the synonyms issue. This is especially important, as this usage is present throughout the paper, starting from the abstract, and to almost every section of the paper.

The case study mentions building sizes and construction, but the location itself is left unclear. It would be of importance, as the location will provide climate information that will also directly impact the retrofit effectiveness. (Somewhere along the case-study, the authors mention Apulia, but it is not clear whether this is the location of the buildings, as it is not mentioned in the initial description of the case study)

In Section 4.1, the authors mention that the retrofitted buildings use 300L boilers for domestic hot water. Are these gas boilers or electric boilers? This is important as in Section 4.2, within the 75m2 unit, the authors mention that the retrofitted building does not use gas/methane. This raises two questions:
1. Is the 75m2 unit no longer in the building described in Section 4.1?
2a. If it is, are the boilers in question are electric, please mention that, it will really reduce confusion. If they are indeed gas boilers, then Section 4.2 will need some reworking.
2b. If it is not part of the building, where is the 75m2 unit located? And why bring a different building/unit to the study?

In the case study itself, it is curious, that the authors bring up retrofit operational and market value improvements, but then, don't look at the medium and long-term impacts of the retrofit. The monetary value and return-on-investment assessments are only complete 5-10 years after the building has been retrofitted and not within the first two years alone, especially if you look at it from the user/occupant/purchaser's perspective. Also, maintenance and operating costs over 5-10 years will reveal more about the effectiveness of the retrofit itself.

That is currently missing in the assessment.

Finally, there is little clarity on how replicable this assessment is for other parts of Europe. If it is not replicable, that raises the question of, what the meaning of this assessment is (especially given the historical-cultural peculiarities as the authors themselves have mentioned)? What do we learn that can be used by the larger European community?

The authors really need to re-structure this paper to make it more tangible and informative to the scientific community and the policymakers at large.

Comments on the Quality of English Language

As mentioned in passing in the comments and suggestions, it is imperative that Sections 1-3 are re-written to improve comprehension.

I would advise liberal use of bullets and numbering when you have multiple points to make, especially, in the Research and Contribution section.

Please get the entire paper proofread by a native English speaker. This is because it almost feels like the authors are translating (I am assuming, from Italian to English) the sentences in a very convoluted way which is unnatural to English. This makes it very difficult to understand.

Reviewer 3 Report

Comments and Suggestions for Authors

The paper is excessively divided into numerous chapters. The section with the most subdivisions appears to be the case studies. It seems that chapters 4.2, 4.2.1, and 4.2.2 may not be necessary and could potentially be consolidated into a single chapter.

Chapter 4.3.2. "a 10% deduction was applied": what is the source from which this information was taken?

The section discussing the transition of case studies from class F to class A and F is overly complex and may be confusing for readers. It would be beneficial to shorten this section to enhance readability and comprehension.

Table six should provide incremental descriptions for construction costs, energy costs, and market values. This will help readers better understand the "monetary convenience" aspect, ensuring clarity.

Understanding the cost estimates provided in the paper is challenging. In the first case, where new construction is considered, the referenced cost data from 2020-21, amounting to 1,300 EUR/sqm, may not be directly comparable to costs calculated for 2023 in the case of upgrading to class E9. Given that construction costs in Italy increased by 17-20% in the two years following the COVID-19 pandemic, it's crucial to account for this difference. It's unclear whether the costs for 2021 are up-to-date.

Additionally, the construction costs for energy retrofits do not include demolition or removal costs, such as those associated with boiler replacements. It's important to acknowledge that the replacement of installations often entails the removal of the previous ones, a factor that has not been addressed in the current analysis.

The expenses related to technical aspects are also not considered: even though the intervention does not need any building permit. there may be costs associated with hiring a technician (architect or engineer) to submit documentation for any other technical or legal compliance (e.g., under Law 10).

To provide a comprehensive analysis, the paper should also include consumption differentials for the various scenarios considered (e.g., class F, class E after renovation, class A after energy retrofit and new construction). This information would help readers understand the potential energy savings associated with each scenario.

Comments on the Quality of English Language

The first issue with the paper is its poor language quality, which requires revision by a native speaker. There are two main problems:

1.         Sentence structure: Many sentences are overly long and filled with subordinates, likely stemming from a literal translation from Italian. This makes them difficult to read and comprehend. For example: “The developer of the intervention was already in possession of the area, so there are no acquisition costs”, which could be written as “Since the developer already owned the area, there were no acquisition costs incurred.

2.         Inappropriate or incorrect terms: Some terms are directly translated from Italian but are not suitable or do not exist in English. For instance, "convenience" is not commonly used in the context it's used in the paper; "feasibility" or "effectiveness" might be more appropriate or any other economic related words. “Requalification intervention” is not used either. Additionally, for terms like "sell on paper" might be better to use "off-plan sales" or "off-plan purchase," and "Analytics estimation" should be replaced with "bill of quantities." Similarly, "amortisation schedule" could be intended as "mortgage repayment plan," among others forms. “Sure return” is not appropriate. Many other examples like that could be found in the paper. These issues need to be addressed to improve the clarity and accuracy of the paper.

Round 2

Reviewer 1 Report

Comments and Suggestions for Authors

I am positive about the article, and have no specific manuscript comments for the editor or authors.

Author Response

The article has been entirely reviewed by a native speaker, with particular reference to Sections 1-2-3, trying to "deItalianize" the structure. In the current form core concepts are immediately discernible.

Reviewer 2 Report

Comments and Suggestions for Authors

The paper and the methodology is interesting, but the paper still needs so much work in terms of language.

The authors have made some corrections and clarifications that help with the technical understanding of the paper. Section 4 is now in good shape (captions of the table could be more detailed, but that is a minor nitpick).

However, the authors have made the bare minimum effort required to improve the language within the rest of the paper. Instead, they have spent the majority of their energy in defending their work and rebutting the points raised in the previous round of review, rather than improving the language and the general quality of the paper.

Comments on the Quality of English Language

I absolutely disagree with the authors' response on the proofreading. The authors responded that "The paper has been reviewed by a native English speaker in order to improve the readability and the clarity of the work."

Not a single native English speaker that I know of would agree that the sentence "In particular, respectively the first component (operational carbon) regards the emissions of CO2 and other greenhouse gases
produced during the use and management of a building." and similar sentences littered throughout the manuscript are grammatically correct or comprehensible.

Then there are sentences such as: "The present paper concerns the aforementioned topic."(first line of Research Objective section) a line that makes it look like the authors are trying to make a word count, and " It requires a complex process that underlies a change of perspective towards a systemic approach able to reduce the carbon emissions associated with the entire life cycle of the building." (Lines 41-42 of manuscript)which are needlessly verbose. It would be more comprehensible if they make a little effort. The latter sentence, if written as: "This is a complex task that requires adoption of a systemic approach that would help reduce the carbon emissions associated with the entire life cycle of the building" makes it shorter and easier to understand.

Section 2, 3, and 4 have sentences which shift tenses mid-sentence. (e.g., Line 149-152). Punctuation marks and commas are missing in some of these long trailing sentences, which does not help matters. (e.g., Lines 185-189)

This is frustrating, because the authors' lackadaisical and slapdash approach to improving the manuscript language, is dragging down the general quality of the paper.

Author Response

(The authors gave the same response as above.)

Reviewer 3 Report

Comments and Suggestions for Authors

The paper could indeed be published in its current form.

Author Response

(The authors gave the same response as above.)

Round 3

Reviewer 2 Report

Comments and Suggestions for Authors

The current version of the paper shows significant improvement, there are still quite a few minor changes needed (Although some of the errors might be either a formatting or a pdf creation issue). The content is much more comprehensible and is getting closer to being ready for publication.

Content-wise, Section 4 is well written. Section 1-3 still needs some work, but it is only minor now.

Of all the things that needs addressing, the most egregious is the beginning of Section 2: "This study is concerned with the topic referred to above. With regards to the building sector and the entire cycle of construction, operational life and decommissioning of buildings, the aim of this paper is to highlight the numerous advantages connected to the use of energy retrofit interventions on existing building stock."

It is one of the more confounding starts for a section I have read in my professional life. I would personally suggest something along the lines of: "This study aims to study the potential changes associated with energy retrofit interventions on existing building stock, with special focus on the economic impact  of said improvement of the building stock (in terms of demolition/retrofit/reconstruction costs, operational cost reductions and property value improvement). The analysis takes into account the entire cycle of construction, operational life and decommissioning of buildings." (The authors can paraphrase or adapt it, but it would be good to not start a section with a reference to the previous section)

The strangest changes noticed (which I see as either unfortunate oversight issues or software/formatting issues when creating the PDF) are in Section 4. There are now some issues with formulas being overwritten by the body of the paper and some of the Table references on line 380/381 and line 410/411 just being empty brackets.

There are a few hanging sentences, which have no end like on Line 324-325. Also, there are stray words and parantheses like "b parties" on line 238 and a "]" on line 239.

The entire paragraph within lines 270-275 is one long sentence that is difficult to comprehend. Breaking it down to smaller sentences would help this paragraph a lot.

The rest of the issues are listed in the language comment section.

Comments on the Quality of English Language

Section 1-3 has some strange word choices and sentence structures, even though it has improved significantly. The use of "underlies" on line 40, where "necessitates/calls for" would be better suited.

Capitalized words in the middle of sentences like "Authors" where they are not required. There are some strange English words like "renovational" instead of "renovation" on line 85.

There are some misplaced punctuation marks, like a missing comma on line 184 and an extra dot on line 198.

Author Response

Dear Reviewer, all the suggestions and required changes have been implemented (although some errors are due to a formatting issue).

Thanks for your valuable comments.